*Brief Communication*
**Co-seismic displacement on October 26 and 30, 2016 ($M_w$ 5.9 and 6.5) -**
**earthquakes in central Italy from the analysis of a local GNSS network**
De Guidi Giorgio[1,2], Vecchio Alessia[1], Brighenti Fabio[1], Caputo Riccardo[3,4,5], Carnemolla Francesco[1], Di Pietro
Adriano[1], Lupo Marco[1], Maggini Massimiliano[3,5], Marchese Salvatore[1], Messina Danilo[1], Monaco
Carmelo[1,2],Naso Salvatore[1]

9            1) Department of Biological, Geological and Environmental Sciences, University of Catania, Italy
10           2) CRUST, UR-UniCT, Catania, Italy
11           3) Department of Physics and Earth Sciences, University of Ferrara, Italy
12           4) Research and Teaching Center for Earthquake Geology, Tyrnavos, Greece
13           5) CRUST, UR-UniFE, Ferrara, Italy

*Corresponding author:* G. De Guidi (deguidi@unict.it)

**1 - Abstract**
On August 24[th] 2016 a strong earthquake ($M_w$ = 6.0) affected Central Italy and an intense seismic sequence.
Field observations, DInSAR analyses, preliminary focal mechanisms as well as the distribution of aftershocks
suggested the reactivation of the northern sector of the Laga Fault, whose southern sector was already
rebooted during the 2009 L'Aquila sequence, and of the southern segment of the Monte Vettore Fault System
(MVFS). Based on these preliminary information and following the stress-triggering concept (Stein et al.,
1999; Steacy et al., 2005), we tentatively identified a potential fault zone most vulnerable to future seismic
events just north of the earlier epicentral area. Accordingly, we planned a local geodetic network consisting
of five new GNSS (Global Navigation Satellite System) stations located at few km of distance on both sides of
the MVFS. This was devoted to picture out, at least partially but in some detail, the possible northward
propagation of the crustal network ruptures. The building of the stations and a first set of measurements
were carried out during a first campaign (September 30[th]-October 2[nd], 2016). On October 26[th] 2016,
immediately north of the epicentral area of the August 24[th] event, a another earthquake ($M_w$ = 5.9) indeed
occurred, followed four days later (October 30[th]) by the mainshock ($M_w$ = 6.5) of the whole 2016 Summer-
Autumn seismic sequence. Our local geodetic network was fully affected by the new events and therefore
we performed a second campaign soon after (November 11[th]-13[th], 2016). In this brief note, we provide the
results of our geodetic measurements that registered the co-seismic and immediately post-seismic
deformation of the two major October shocks documenting in some detail the surface deformation close to
the fault trace. We also compare our results with the available surface deformation field of the broader area,
obtained on the basis of the DInSAR technique, and show an overall good fit.

**2 - Geological framework**
The central Apennines are characterized by northeast-verging thrust-propagation folds, involving Mesozoic-
Tertiary sedimentary successions. During the 2016 sequence, coseismic deformation was recorded at the
rear of the Sibillini Thrust that separates the homonymous mountain chain from the Marche-Abruzzi foothills
(Fig. 1). According to many studies in the area, the main thrust-related anticlines and associated reverse faults
have been dissected and/or inverted by NNW-SSE trending Quaternary normal and oblique-slip faults (Figs.
1 and 2), in particular by the Norcia Fault System (NFS) (Calamita and Pizzi, 1992; Calamita et al., 1982; 1995;
1999; 2000; Blumetti et al., 1990; Blumetti, 1995; Brozzetti and Lavecchia, 1994; Cello et al., 1998; Galadini
and Galli, 2000; Pizzi and Scisciani, 2000; Pizzi et al., 2002; Boncio et al., 2004 Galadini, 2006; Gori et al., 2007)
and the Mt. Vettore Fault System (MVFS) (Calamita and Pizzi, 1991; Coltorti and Farabollini, 1995; Cello et
al., 1997; Pizzi et al., 2002; Galadini and Galli, 2003; Pizzi and Galadini, 2009) (Figs. 1 and 2). Conversely,
Pierantoni et al. (2013) suggest that the major Mt. Sibillini Thrust has not been yet dissected by quaternary
normal faulting, though some fresh morphological scarps with free faces in the carbonate bedrock and/or
affecting recent slope deposits have been observed and attributed to the local seismic activity.
Within a distance of few tens of kilometers, large evidence ground deformation has been provided by several
recent earthquakes, like the 1979 Norcia event ($M_w$ 5.9, reactivating the Norcia Fault; e.g. Deschamps et al.,
2000), the 1984 Gubbio ($M_w$ 5.6, Gubbio Fault; e.g. Boncio et al., 2004), the 1997 Colfiorito ones ($M_w$ 5.7, 6.0
and  5.6, Calfiorito-Cesi-Costa fault system; e.g. Cello et al., 1997), the 2009 L'Aquila mainshock and the
Campotosto aftershock ($M_w$ 6.3 and 5.4, Upper Aterno Valley-Paganica fault sytsem and Gorzano Fault;
Blumetti et al. 2013) and basically the same occurred with the 2016 seismic sequence.
Surface evidence of the August 24[th] (e.g., EMERGEO WG, 2016; Livio et al., 2016; Aringoli et al., 2016) was
mainly observed in the area of the Laga basin (Gorzano Fault), which corresponds to the footwall block of
Sibillini Thrust, while debated ground ruptures (e.g. Valensise et al., 2016) also occurred in the southern
sector of the MVFS, which belongs to the hanging-wall block of the orogenic structure. In contrast, as a
consequence of the mainshock of October 30[th], the entire western flank of the Monte Vettore was affected
by impressive geological effects and clear coseismic ruptures mapped for a minimum length of 15 km,
between the Castelluccio di Norcia and Ussita (EMERGEO WG, 2016) (Fig. 2). The surface ruptures occured
along distinct fault splays of the fracture system. For example, along the western slope of  Monte Vettore
three main west dipping splays were activated together with two antithetic branches (Figs. 1 and 2). The
observed vertical offset reached 2 m along the main west dipping fault segment, where the slickensides show
a prevalent dip-slip component of motion. Vertical displacements of a few centimetres were also recorded
along an antithetic surface rupture bordering to the west the Castelluccio plain, about 6-7 km far from the
main ground rupture, possibly connected to a secondary fault (Figs. 1B and 2).
It is worth to note that the August-October earthquakes occurred in a sector of the Central Apennines
characterized by high geodetic strain-rates (e.g., Devoti et al., 2011; D'Agostino 2014), where several
continuous GNSS stations are operating.

**Implementation and Analysis of UNICT discrete GPS stations**
Following the August 24[th], $M_w$ 6.0 earthquake, the GEOmatic Working Group of the Catania University
(UNICT) in collaboration with the SpinOff EcoStat s.r.l. and researches of the Ferrara University, started a
detailed monitoring of ground deformation in the epicentral area using the Global Navigation Satellite System
(GNSS) technique. The GNSS measurement has been made in static mode, setting the time at 6 hours and
post-processing position, in order to reduce tropospheric error and using IGS precise products for orbits. The
IGS station coordinates were kept fixed in order to align the final velocity field with the WGS84 reference
frame. The measurement mode, adopted for receiver-satellite range determination, is performed with a
double frequency receiver, allowing phase and code measurements on the signal carrier (L1, L2, C1, P1, P2,
S1, S2). The coordinates estimation is based on the principle of minimum squares.
For this aim, five GNSS stations have been installed on new benchmarks purposely built by the working group
and here referred to as UNICT network (Fig. 3). These new stations have been realized taking into account
the following criteria:
I.      the distribution of the existing permanent and discrete measurement benchmarks belonging to
different networks that were active before the event of 24 August (IGM; RING; CAGEONET; DPC;
ISPRA) (Fig. 2B).
II.      the seismotectonic setting of the area in relation to the macroseismic data and to the reactivated
structures (Figs. 1 and 2);
III.      surface and deep geometry of the major faults related to tectonic setting (Fig. 1B).
IV.       the lack of possible gravitational instabilities in both static and dynamic conditions in sites where the
new benchmarks are built.

Based on the above criteria, the working group installed the benchmarks at the bottom of both western and
eastern slopes of Mt. Vettore, within an area about 8 km-long and 5 km-wide in the N-S and E-W directions,
respectively. The distribution of the benchmarks was planned for reconstructing the principal deformation
zone developed as a consequence of the August 24th event (Fig. 2) and particularly with points:
I.      much closer to the epicentral area than the already existing ones belonging to other networks   (Fig.
2B);
II.      characterized by equivalent distances from the reactivated Mt. Vettore Fault segments (Fig. 2);
III.      within a distance of 30 km from the closest permanent network points that have been not affected
by deformation, therefore allowing a rigorous elaboration during the post processing phases.

The building of GNSS monument on the UNICT benchmarks consists of the following steps (Fig. 3):
I.      selection of a suitable site, corresponding to a massive rocky outcrop or a man-made monument with
foundation; these sites must be also free of structures or other natural elements in the surroundings
that may constitute a perturbation during recording;
II.      testing of GPS signal reception by short-term exams, and control of parameters set through the
quality   check   carried   out   by   software   TEQC   (http://www.unavco.org/software/data-
processing/teqc/teqc.html);
III.      implementation of the hole for housing the bushing and check of its verticality; the hole has a
diameter of 35 mm and a depth of 100 mm, it is realized through small-sized battery-powered
equipment (Makita DHR243 hammer drill);
IV.       fixing and anchoring of the knurled steel bushing (length 67 mm and diameter 20 mm), with bi-
component resins or quick-setting cements (Fig. 3);
V.       following the cementation to the artefact or to rocky outcrop, a male-male threaded bar can be
screwed in until end of stroke; the height could be variable and this fact is considered in the data
processing. We have used a threaded bar 670 mm-high.

The GPS monument is thus completed with a GNSS receiver TOPCON, mounting a HiPer V antenna,
characterized by 226 channels and position accuracy with band L1+L2 in Static mode of 3 mm + 0.1 ppm
(horizontal) and 3.5 mm + 0.4 ppm (vertical). All registrations last six hours in static mode.
Following the August 24th event, at the end of September 2016 the working group curried out the first survey
campaign with the installation of five UNICT benchmarks: two stations were located east of the Mt. Vettore
fault (VTE1,VTE2), the other three (VTW3,VTW4, VTW5) west of the fault (Figs. 4 and 5). During November
2016 (*i.e.* after the October 30th event), a second field campaign was carried out following the same
procedure and using the same instrumentation. The second set of measurements allowed us to record the
co-seismic displacement caused by both the $M_w$ 5.9 and $M_w$ 6.5 events of October 26th and 30th, respectively
(doy (day of year) 2016/274 and doy 2016/318).

The data from survey-mode GNSS stations have been downloaded and processed using TOPCON Magnet analysis software evaluating co-seismic solutions and comparing with AUSPOS web-based online services for GPS data processing (Ocalan et al., 2013), whose engine is based on Bernese 5.2 software. In the software TOPCON, the baseline is automatically created for any pair of static occupations, where we set up six hours for Minimum Duration and the baselines max length of 50 km, cut-off angle of 15°, troposphere model Goad-Goodman and, finally, meteo model NRLMSISE (neutral temperature and densities in Earth's atmosphere). For the analyses we referred to the measurement of a stable reference frame of five GNSS stations belonging to the RING (Rete Integrata Nazionale GPS) network, with a maximum baseline length of 50 km, using stations CESI, GNAL, GUMA, MTER and MTTO (Figs. 4 and 5). Data processing has been carried out with adjustment by Least Squares and a TAU Criterion.

**Concluding remarks**

Using the GNSS technique, we investigated the ground deformation occurred in the surroundings of the Mt. Vettore Fault System during the 2016 central Italy seismic sequence. This foresight action allowed us to record the co-seismic and part of the post-seismic deformation of the second and third (strongest) events (Mw 5.9 and $M_W$ 6.5) on October 26[th] and October 30[th], 2016, respectively. Taking into account the geometry of the fault system in the broader epicentral area and following the stress-triggering concept (Stein et al., 1999; Steacy et al., 2005), we have identified a potential fault zone most vulnerable to future seismic events just north of the fault segment reactivated during the August 24[th] earthquake (Figs. 2B and 5). With this in mind, in order to measure the post seismic deformation and to possibly record the potential migration of the co-seismic process, we selected some sites and built five new GNSS benchmarks, distributed east and west of the northern-central segment of the Mt. Vettore Fault System. For site selection we also considered the presence and distribution of other benchmarks located before the second seismic event by other research groups (IGM; RING; CAGEONET; DPC; ISPRA). The epicentral location of the October events confirmed our guess and then we performed soon after a second campaign of measurements for quantifying the relative motion of the stations.

The measured deformation (with 95% confidence errors) is characterised by both horizontal and vertical movements. In particular, the east benchmark VTE1 recorded 312 mm of eastward horizontal displacement and 29 mm of upward motion, while the VTE2 recorded 282 mm of eastward horizontal displacement and 67 mm of upward component of motion. On the contrary, all three western benchmarks recorded westward horizontal displacements (419, 288 and 26 mm) and subsidence (707, 288 and 769 mm) for stations VTW5, VTW4 and VTW3, respectively. In conclusion, we documented ca. 730 mm of ENE-WSW lengthening on a distance of 7 km in correspondence of the northern sector of the Mt. Vettore Fault Segment, while the off-fault vertical displacement between footwall and hanging-wall blocks was 736 mm.

We also compared our results with the displacement distribution obtained by other research group with DInSAR techniques, recorded between October 26th 2016 (pre-event images) and November 1st 2016 (post-event images), and other GNSS stations, active before the second seismic event. In Fig. 5 we may observe the overall consistency of the different approaches and datasets.

**Acknowledgments** This paper was carried out with the financial support of the University of Catania (FIR 2014 Project Code 2C7D79, Scientific Supervisor: G. De Guidi) and University Spin Off of Catania EcoStat s.r.l.

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

agosto.html; Sequenza sismica di Amatrice, Norcia, Visso: approfondimenti e report scientifici

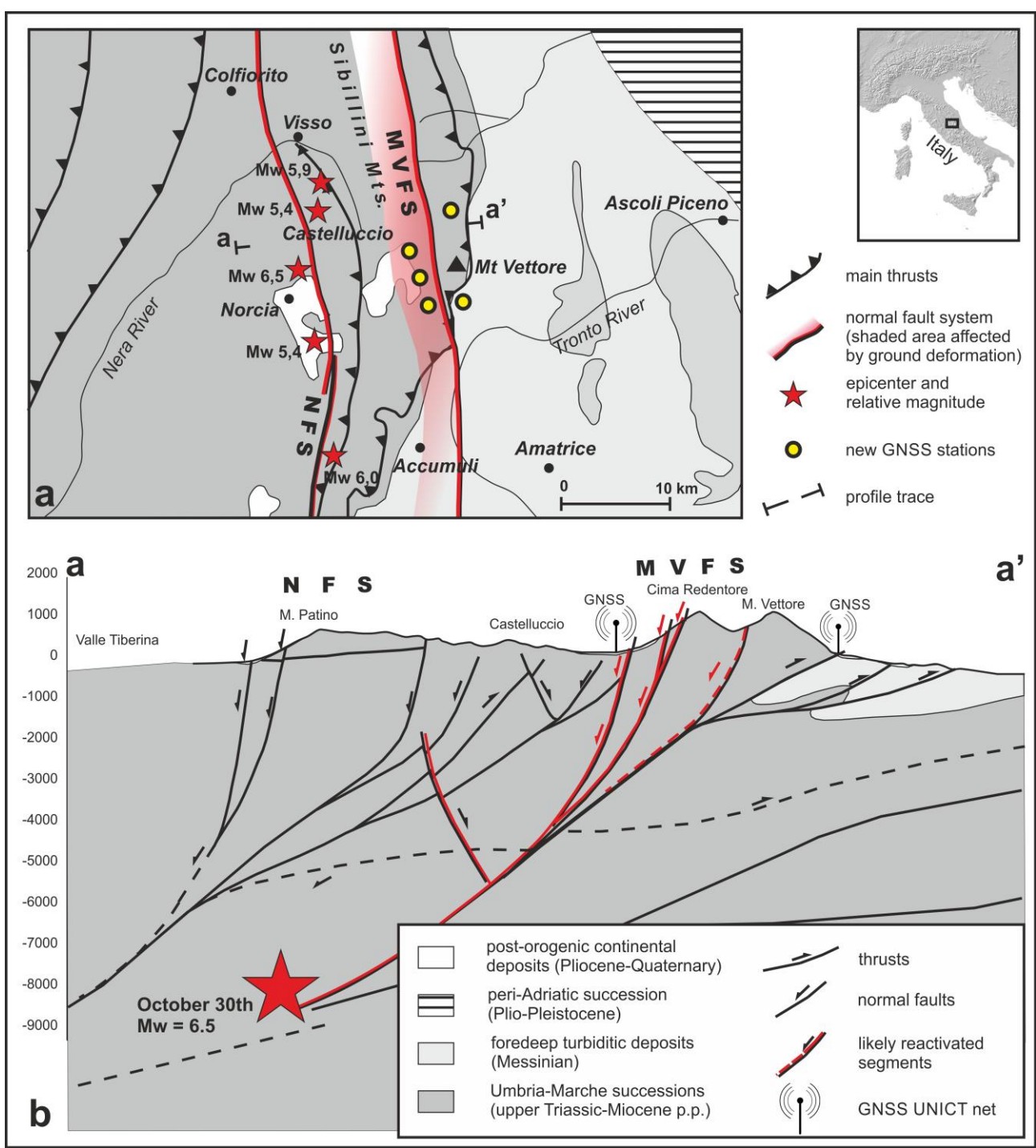

Fig. 1 - Simplified seismotectonic map of central Apennines (A) and geological profile across the epicentral
area (B). The location of the major event (October 30th) is from GdL INGV (2016), while the main
geostructural features from Pierantoni et al. (2013) and Mantovani et al. (2011) modified).

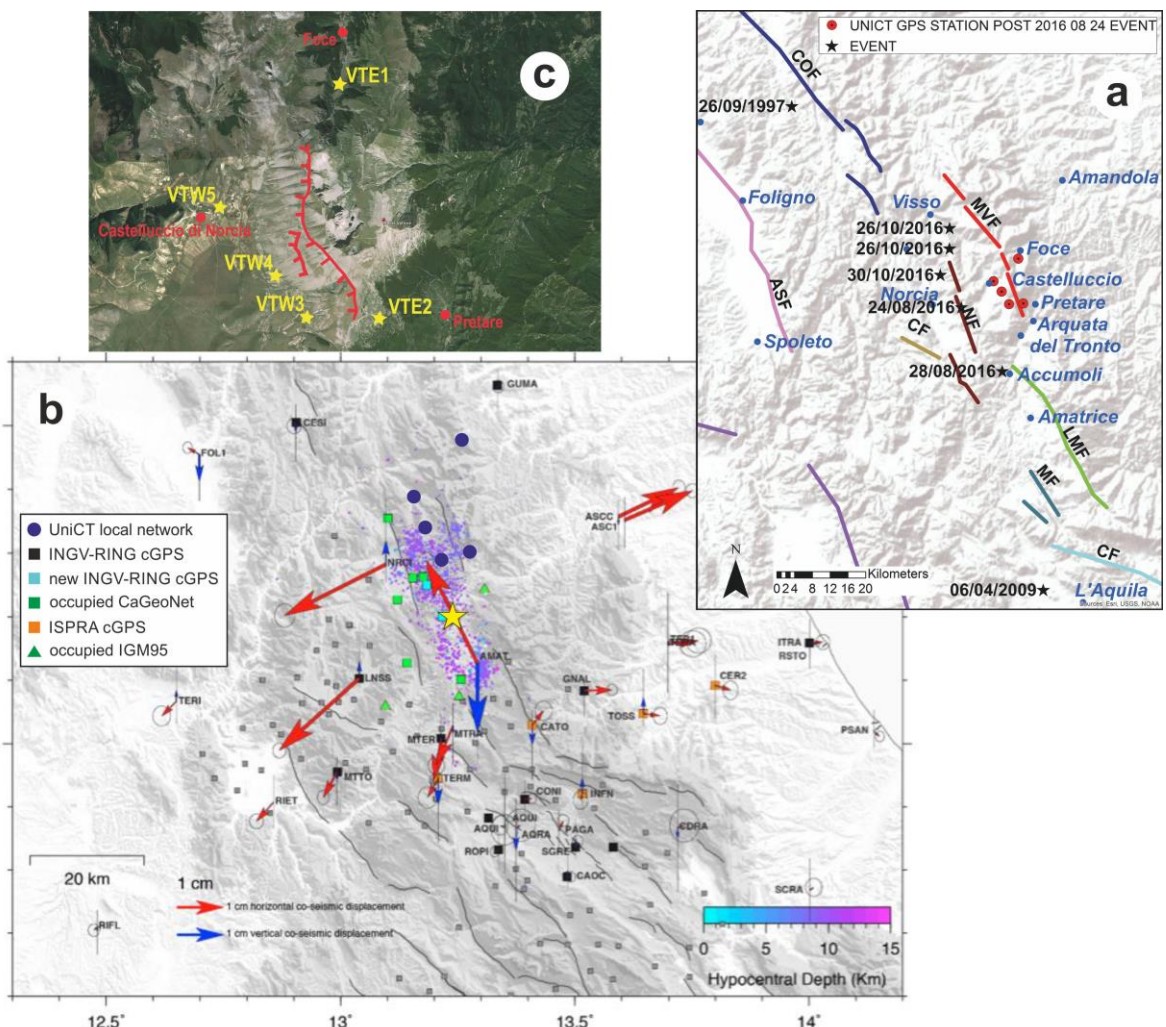

Fig. 2 – a) Digital Elevation Model with shaded relief of central Apennines showing the active fault system
and the major events since 1997 (ASF: Assisi Fault; COF: Colfiorito Fault; CF: Cascia Fault; MVF: Mt. Vettore
Fault; NF: Norcia Fault; LMF: Laga Mts. Fault; MF: faults of the Montereale basin). b) Horizontal (red arrows)
and vertical (blue arrows) consensus co-seismic displacements (with 68% confidence errors), and the local
UniCT GPS network. The aftershocks of the August 24th, Mw 6.0 main event (yellow star) are colored as a
function of depth (from http://iside.rm.ingv.it); c). GoogleEarth map showing the new five GNSS stations
(yellow stars) located in the near field of (and surrounding) the October 30th coseismic ground ruptures (red
lines).

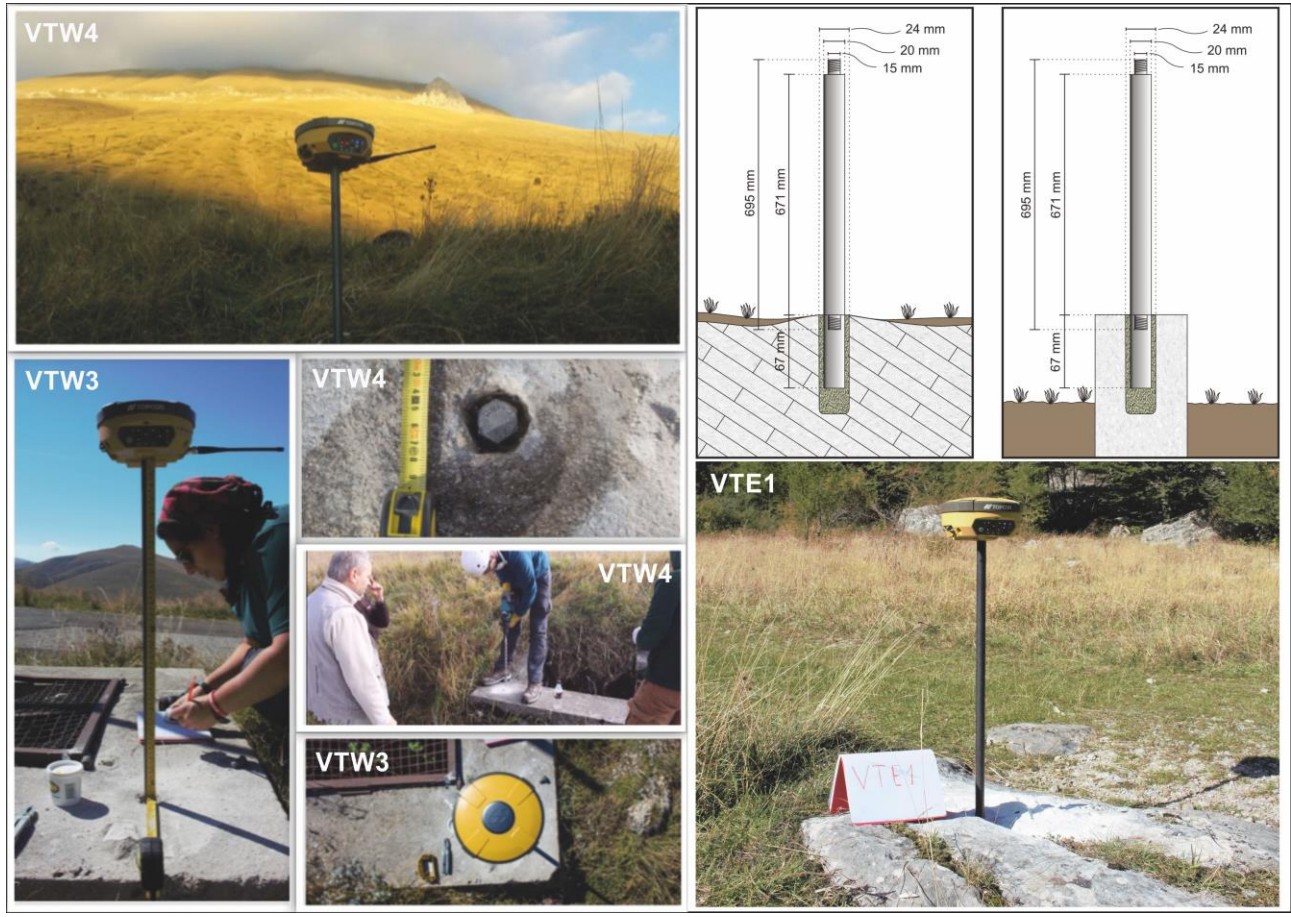

Fig. 3 - Synoptic picture showing installation of the new GNSS stations, measurement and processing phases.

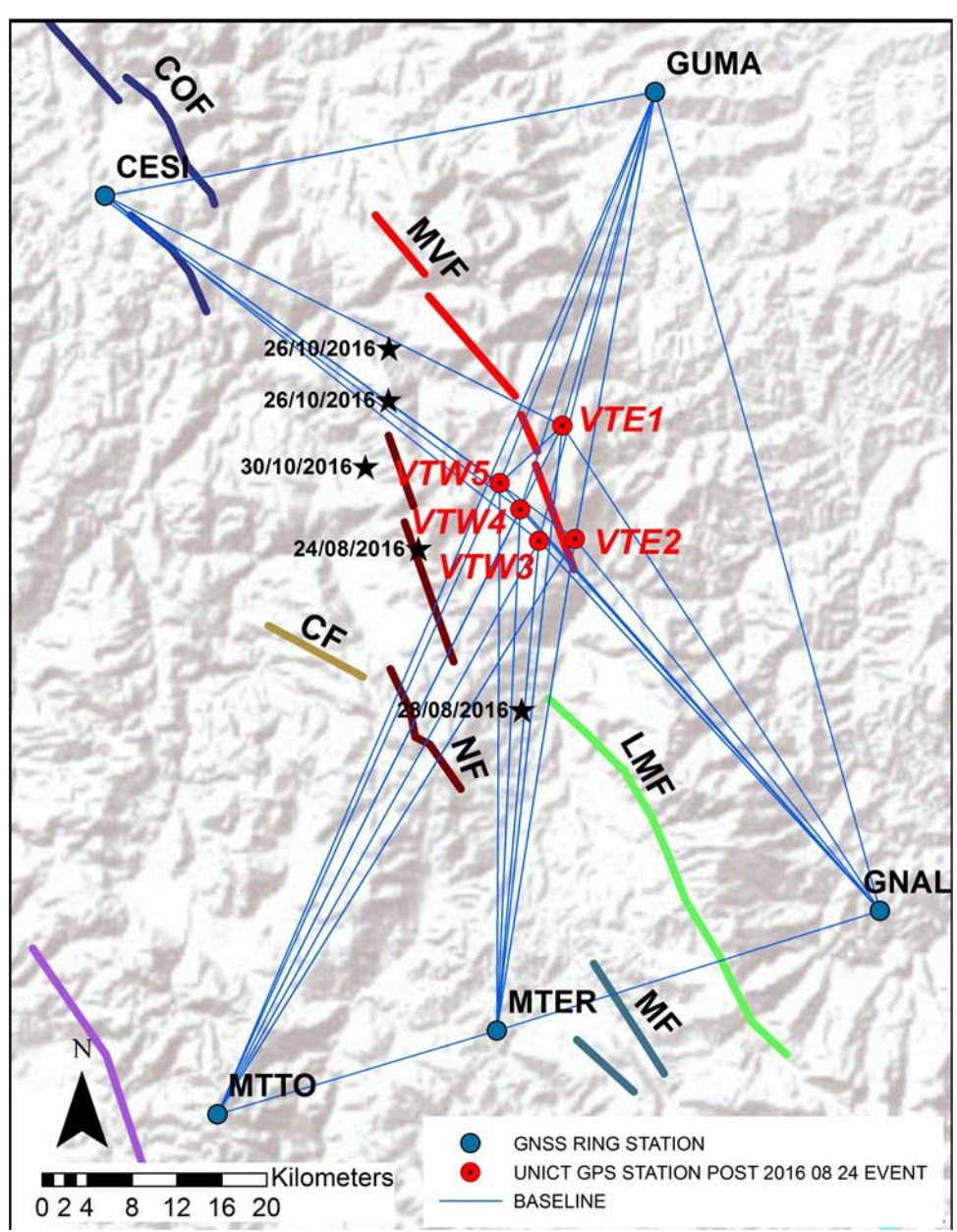

Fig. 4 - Baselines obtained by combining the new GPS UNICT stations with selected GNSS ones from the RING Network.

Fig. 5 - Color-coded maps showing the E-W (a) and vertical (b) displacement distribution obtained by the DInSAR technique (http://www.irea.cnr.it/index.php?option=com_k2&view=item&id=761:nuovi-risultati-sul-terremoto-del-30-ottobre-2016-ottenuti-dai-radar-dei-satelliti-sentinel-1) recorded On October 26th 2016 (pre-event images) and on November 1st 2016 (post-event images).The red and blu arrows represent the consensus pre-, co-, and post-seismic displacements (with 95% confidence errors) on the basis of the GNSS UNICT network. Epicenters of major shocks are from http://ring.gm.ingv.it.

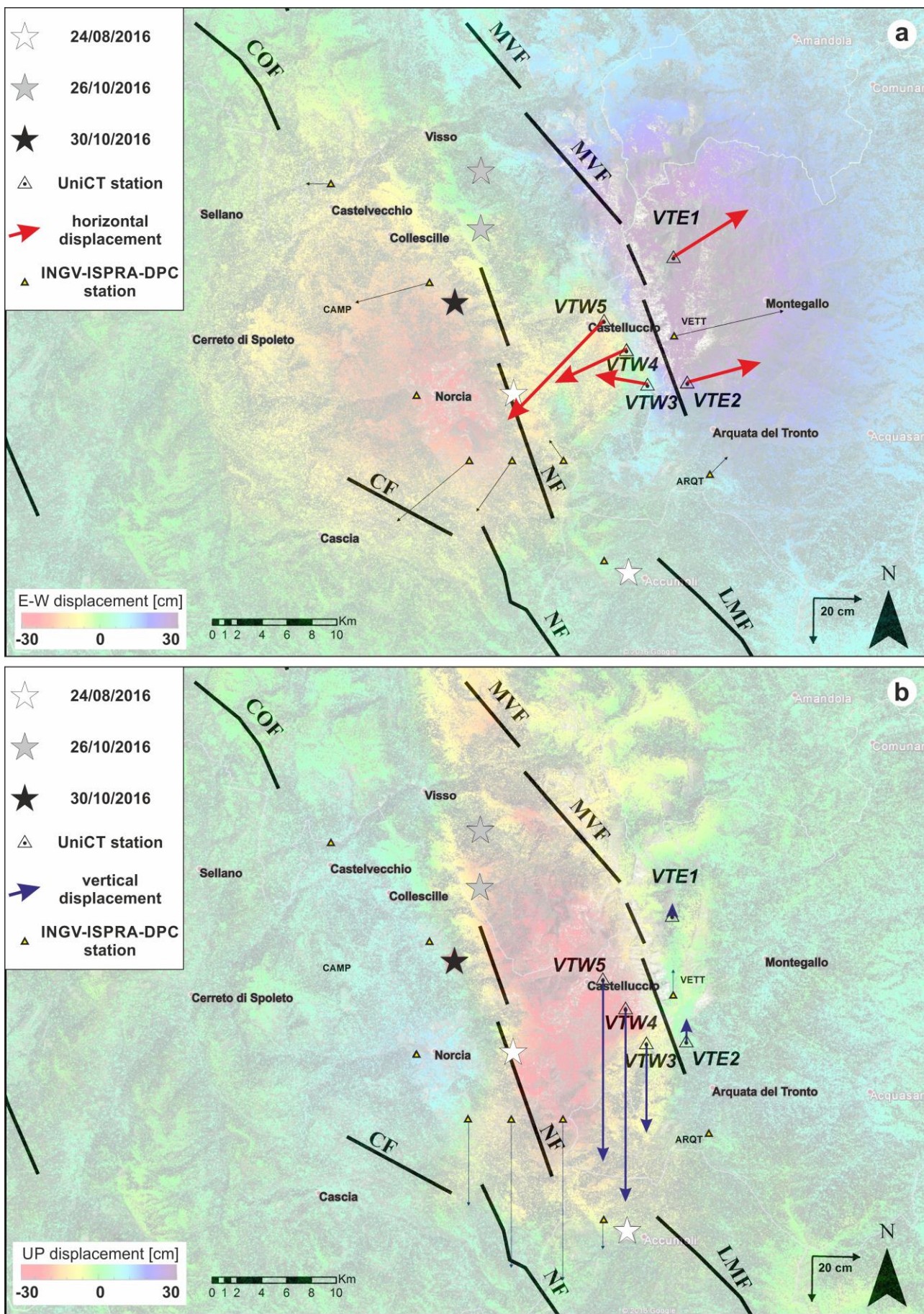


| ID | Station | Longitudine | Latitudine | disp$_{N-S}$ | disp$_{E-W}$ | disp$_{UP}$ | unc$_{N-S}$ | unc$_{E-W}$ | unc$_{UP}$ |
|---|---|---|---|---|---|---|---|---|---|
| VTE1 | FOCE_SENTIERO | 13° 15' 57,45166'' | 42° 51' 57,04340'' | 141 | 312 | 29 | 15.5 | 16.5 | 44.0 |
| VTE2 | PRETARE | 13° 16' 33,20959'' | 42° 47' 56,56780'' | 60 | 282 | 67 | 19.0 | 16.5 | 46.0 |
| VTW3 | QUARTUCCIOLO | 13° 14' 46,41153'' | 42° 47' 56,57032'' | 198 | 26 | -349 | 15.5 | 14.5 | 36.0 |
| VTW4 | COLLE_CURINA | 13° 13' 55,01245'' | 42° 48' 59,62491'' | 102 | 288 | -769 | 15.5 | 15.0 | 36.0 |
| VTW5 | CASTELLUCCIO_VALLE | 13° 12' 56,20423'' | 42° 49' 54,89014'' | 353 | 418 | -707 | 15.0 | 13.5 | 37.5 |


Tab 1 - Three components co-seismic displacements and relative uncertainties estimated for the GNSS
stations of the UNICT network. Coordinates are WGS84 east and north, respectively. All displacement and
uncertainty values are in millimeters. For all stations, the cut-off angle is 15°, the troposphere model is the
Goad-Goodmar and the meteo model used is NRLMSISE. The table can be download as ASCII file on the
INGVRING web page (http://ring.gm.ingv.it).