# Peer review of "Brief Communication"

_Natural Hazards and Earth System Sciences, 2017_

## Referee Comment (RC1) · A. Ganas (Referee) · 7 May 2017

The authors present the results of two GNSS campaigns in central Italy, close to the epicentre area of the 2016 seismic sequence. The work is not clearly documented so I recommend major revision. Please see my comments below.

General comments:

a) The authors need to provide more details on GNSS data processing. They report that they used a commercial software but this is not enough. In addition, the results are not clearly presented. So I suggest to redo Table 1 with re-

**[NHESSD](NHESSD)**

Interactive
comment

sults: N-S displacement, E-W displacement, Up displacement, N-S uncertainty, E-W uncertainty Up uncertainty. Also, please report the sampling interval of the geodetic observations and the cut-off angle of observations. See articles such as http://www.annalsofgeophysics.eu/index.php/annals/article/view/6418/6508 for data processing of local GNSS networks.

b) the authors neglect recent Italian literature on the 2016 seismic sequence http://www.annalsofgeophysics.eu/index.php/annals/issue/view/515

c) The comparison with SAR interferometry results shows notable differences in the horizontal component (compare red colour extent in fig. 4A with vectors of stations VTW3 etc.). What is the exact day/hour of the pair of Sentinel images, is it after the Oct. 26, 2016 event or before? So the dInSAR image shows the deformation of 1 or 2 events?

d) The authors report cumulative GNSS results from 2 earthquakes, October 26 and October 30, 2016. This needs to be very clear and should be included in abstract, conclusions.

e) the addition of horizontal and vertical displacements along the fault is wrong as the GNSS measurements are point measurements and are valid for the particular site only.

f) The discussion in Fig. 5 is vague. The authors should expand more on what they believe is worthy for more investigation. Also report more on the data presented in the east-west transect.

g) The manuscript figures need re-organisation so the material is clearly presented.

Fig.1: increase size of map (1A). remove empty (white space) in fig. 1B. It is uncertain to put the hypocentre of the M6.5 event on this cross-section given a) the depth - location uncertainty and b) the lack of association with surface faulting in this section. So make 1A larger and put 1b below. Also. in 1B show seismic faults with red lines. In fig. 1A show the GNSS points which you refer in this paper. In addition, the concentric

[Figure]

circles indicate seismic wave propagation? I suggest that this also be left out as this is not the focus of this paper.

Fig. 2: insert a box in 2A showing the extent of 2C.

Fig. 3: indicate which GNSS station (e.g. VTE1? VTE2? etc.) is shown in the field photographs. I suggest to insert a new figure expanding the material of Fig.3 (lower right) showing better station baselines, faults, epicentres of October 26 and October 30, 2016 earthquakes and co-seismic ruptures. This will be a key figure to help readers to understand the relation between GNSS station location, earthquake effects etc.

Fig.4; there are five grey stars in this figure. Please leave only the 3 mainshocks and represent them with red colour. Clarify that InSAR scale bar is in cm.

Specific comments: line 3: replace discrete with campaign line 17: please indicate which agency provided moment magnitude estimation line 19: please indicate which agency provided moment magnitude estimation line 34: replace doy with DOY line 136: replace registrations with recordings

Athens, 7 May 2017 Athanassios Ganas

---

## Referee Comment (RC2) · J. Browning (Referee) · 23 May 2017

This manuscript presents ground deformation data associated with a series of earthquakes in October 2016 in the Central Apennines region of Italy. The researchers placed an array of GNNS stations within a fault zone north of the site of the August 24th, 2016 rupture and fortuitously captured the co-seismic deformation of two later earthquake ruptures in October 2016. The GNSS displacement data is supported by DInSAR observations and appears to indicate the complex activation of a thrust fault to the East and a normal fault to the West of the main MVF fault segment. I believe that

the data is robust, important and timely and therefore recommend that this brief communication is suitable for publication in Natural Hazards and Earth Systems Science following some minor revision.

I have a few main suggestions that I hope will be considered to improve the manuscript as follows: Although there is a concluding remarks section it is difficult to ascertain clearly the main conclusions from this study. Just adding a few sentences after lines 186 where the majority of the displacement data is presented would be good, you might just simply explain the geological significance and structures observed from that data. The discussion regarding the 'seismic efficiency' of the Norcia fault system at lines 191-192 is somewhat ambiguous as it is currently presented, can you elaborate on this please?

The quality of Figure 5 is low, in my opinion it is not at the standard of an international publication. The overall look of the figure is poor, the x-axis is missing a label and the legend displays 'serie 1' which is not helpful information. Furthermore, there is very little annotation of, for example, the dashed red lines, and so interpreting the figure is almost impossible.

There are some inconsistencies regarding the format of date and time throughout the manuscript, for example, there are at least 3 different date formats used in the abstract alone. At lines 16-17 the format is month and day (full words and with the ordinal indicator), then line 24 uses month and day without an ordinal indicator. Line 28 uses day, month, year (full words). Line 31 uses day-month, year (numeric format). Then finally, line 34 uses the format day of year (doy) format. I think it would be better to pick one format and use that consistently throughout.

As well as main comments above I also list a series of minor points as follows:

Abstract:

Line 18 – add 'the' – '….boundary between the Marche and Umbria regions…'

Line 21 – omit 'has been' and replace with 'was' – '......deformation was recorded at the rear...' Line 28 – add 'the' – '.....hypocenter of the major event...'

Line 29 – replace 'points' with 'sensors' or 'stations'

Line 30 – Define GNSS

Line 38 – Please check the format requirements for entering long hyperlinks. I am not sure about this but in the current form this looks a bit cumbersome.

Active faults:

Line 50 – replace 'authors' with 'studies'

Line 51 – The choice of the word 'characterizing' here makes the sentence and grammar unclear, consider rephrasing this.

Line 83 - Add 'a' – 'Vertical displacement of a few centimetres....'

Implementation and analysis

Lines 135-136 – Incorrect using of comma and decimal place. 0,1 ppm, 3,5 ppm etc should be 0.1 ppm and 3.5 ppm etc.

Line 138 – replace 'have been' to 'were'

Line 141 – replace 'to record the' with 'the recording of'

Table 1: Can the abbreviations be written in full? The columns of Height, Ground Distance (elevation?) and Delta Ell. use the incorrect decimal point and comma format as previously discussed, please correct.

Concluding remarks

Line 170 – Replace 'based on' with 'Using' and omit 'a' and 'the' and add 'which' omit the part about partial reactivation (or reword the sentence because the grammar is unclear)– 'Using the GNSS technique detailed monitoring of ground deformation, which

occurred in the Mt. Vettore Fault segment, has been carried out.'

Line 175 – Should coseismic be hyphenated as in the title?

Line 180 – add 'of' – '. . .344 mm of eastward horizontal displacement and 34 mm of upward displacement' etc.

Many thanks, John Browning

---

## Author Comment (AC1) · 13 Jun 2017

This manuscript presents ground deformation data associated with a series of earthquakes in October 2016 in the Central Apennines region of Italy. The researchers placed an array of GNNS stations within a fault zone north of the site of the August 24th, 2016 rupture and fortuitously captured the co-seismic deformation of two later earthquake ruptures in October 2016. The GNSS displacement data is supported by DInSAR observations and appears to indicate the complex activation of a thrust fault to the East and a normal fault to the West of the main MVF fault segment. I believe that

the data is robust, important and timely and therefore recommend that this brief communication is suitable for publication in Natural Hazards and Earth Systems Science following some minor revision.

I have a few main suggestions that I hope will be considered to improve the manuscript as follows: Although there is a concluding remarks section it is difficult to ascertain clearly the main conclusions from this study. Just adding a few sentences after lines 186 where the majority of the displacement data is presented would be good, you might just simply explain the geological significance and structures observed from that data. The discussion regarding the 'seismic efficiency' of the Norcia fault system at lines 191-192 is somewhat ambiguous as it is currently presented, can you elaborate on this please?

Reply: We have simplified the discussions relating the displacement data explaining geological significance, sea also new shapes of the Figure 1. Furthermore we have reorganised the paper's figures. removing the Figure 5 which represented a qualitative indicator that must be supported by new data, and splitting the Figure 3.

The quality of Figure 5 is low, in my opinion it is not at the standard of an international publication. The overall look of the figure is poor, the x-axis is missing a label and the legend displays 'serie 1' which is not helpful information. Furthermore, there is very little annotation of, for example, the dashed red lines, and so interpreting the figure is almost impossible.

Reply: Figure 5 has been removed

There are some inconsistencies regarding the format of date and time throughout the manuscript, for example, there are at least 3 different date formats used in the abstract alone. At lines 16-17 the format is month and day (full words and with the ordinal indicator), then line 24 uses month and day without an ordinal indicator. Line 28 uses day, month, year (full words). Line 31 uses day-month, year (numeric format). Then finally, line 34 uses the format day of year (doy) format. I think it would be better to pick one format and use that consistently throughout.

Reply: We corrected date format, we used an unique format

As well as main comments above I also list a series of minor points as follows:

Abstract:

Line 18 – add 'the' – ': : :.boundary between the Marche and Umbria regions: : :'

Line 21 – omit 'has been' and replace with 'was' – ': : :: : :deformation was recorded at the rear: : :' Line 28 – add 'the' – ': : :..hypocenter of the major event: : :'

Line 29 – replace 'points' with 'sensors' or 'stations'

Line 30 – Define GNSS

Line 38 – Please check the format requirements for entering long hyperlinks. I am not sure about this but in the current form this looks a bit cumbersome.

Reply: we are agree with you but we checked the hyperlink and we found that it is impossible change the format.

Active faults:

Line 50 – replace 'authors' with 'studies'

Line 51 – The choice of the word 'characterizing' here makes the sentence and grammar unclear, consider rephrasing this.

Line 83 - Add 'a' – 'Vertical displacement of a few centimetres: : :.'

Implementation and analysis

Lines 135-136 – Incorrect using of comma and decimal place. 0,1 ppm, 3,5 ppm etc should be 0.1 ppm and 3.5 ppm etc.

Line 138 – replace 'have been' to 'were'
Line 141 – replace 'to record the' with 'the recording of'

Table 1: Can the abbreviations be written in full? The columns of Height, Ground Distance (elevation?) and Delta Ell. use the incorrect decimal point and comma format as previously discussed, please correct. Reply: we completely modified the table 1

Concluding remarks

Line 170 – Replace 'based on' with 'Using' and omit 'a' and 'the' and add 'which' omit the part about partial reactivation (or reword the sentence because the grammar is unclear)– 'Using the GNSS technique detailed monitoring of ground deformation, which occurred in the Mt. Vettore Fault segment, has been carried out.' Line 175 – Should coseismic be hyphenated as in the title? Line 180 – add 'of' – ': : :344 mm of eastward horizontal displacement and 34 mm of upward displacement' etc.

Reply: We are very grateful for your grammatical correction, we followed all your suggestion

Please also note the supplement to this comment:
http://www.nat-hazards-earth-syst-sci-discuss.net/nhess-2017-130/nhess-2017-130-AC1-supplement.pdf

**Supplement:**

[revised manuscript text omitted]

*Fig. 4 Baselines obtained by combining the new GNSS UNICT stations with selected GNSS ones from the RING Network.*

                                                    ------------------------------->

*Fig. 5 Color-coded maps showing the E-W (a) and vertical (b) displacement distribution obtained by the DInSAR technique (http://www.irea.cnr.it/index.php?option=com_k2&view=item&id=761:nuovi-risultati-sul-terremoto-del-30-ottobre-2016-ottenuti-dai-radar-dei-satelliti-sentinel-1) recorded On October 26th 2016 (pre-event images) and on November 1st 2016 (post-event images).The red and blu arrows represent the consensus pre-, co-, and post-seismic displacements (with 95% confidence errors) on the basis of the GNSS UNICT network. Epicenters of major shocks are from http://ring.gm.ingv.it.*

[Figure]

---

## Author Comment (AC2) · 13 Jun 2017

General comments: a) The authors need to provide more details on GNSS data processing. They report that they used a commercial software but this is not enough. In addi-tion, the results are not clearly presented. So I suggest to redo Table 1 with re-sults: N-S displacement, E-W displacement, Up displacement, N-S uncertainty, E-W uncertainty Up uncertainty. Also, please report the sampling interval of the geodetic observations and the cut-off angle of observations. See articles such as http://www.annalsofgeophysics.eu/index.php/annals/article/view/6418/6508 for data

processing of local GNSS networks.

Reply: we created a new table with your suggest

b) the authors neglect recent Italian literature on the 2016 seismic sequence http://www.annalsofgeophysics.eu/index.php/annals/issue/view/515

Reply: we revised the bibliography (line 236-237)

c) The comparison with SAR interferometry results shows notable differences in the horizontal component (compare red colour extent in fig. 4A with vectors of stations VTW3 etc.). What is the exact day/hour of the pair of Sentinel images, is it after the Oct. 26, 2016 event or before? So the dInSAR image shows the deformation of 1 or 2 events?

Reply: We have simplified the discussions relating the displacement data explaining geological significance, sea also new shapes of the Figure 1. Furthermore we have reorganised the paper's figures. removing the Figure 5 which represented a qualitative indicator that must be supported by new data, and splitting the Figure 3. This difference are explained with the difference type of data. The GNSS data are punctual measuring while DInSAR are areal measuring so it is probably that there are some difference between these data. The DInSAR image shows the deformation before and after the two events (line 175-176).

d) The authors report cumulative GNSS results from 2 earthquakes, October 26 and October 30, 2016. This needs to be very clear and should be included in abstract, conclusions.

Reply: Of course we have rewrite the abstract and conclusion considering this suggestion

e) the addition of horizontal and vertical displacements along the fault is wrong as the GNSS measurements are point measurements and are valid for the particular site only.

Reply: We compute again the displacements including new data in the table and conclusion

f) The discussion in Fig. 5 is vague. The authors should expand more on what they believe is worthy for more investigation. Also report more on the data presented in the east-west transect.

Reply: Figure 5 has been removed

g) The manuscript figures need re-organisation so the material is clearly presented.

Fig.1: increase size of map (1A). remove empty (white space) in fig. 1B. It is uncertain to put the hypocentre of the M6.5 event on this cross-section given a) the depth - location uncertainty and b) the lack of association with surface faulting in this section. So make 1A larger and put 1b below. Also. in 1B show seismic faults with red lines. In fig. 1A show the GNSS points which you refer in this paper. In addition, the concentric circles indicate seismic wave propagation? I suggest that this also be left out as this is not the focus of this paper.

Fig. 2: insert a box in 2A showing the extent of 2C.

Fig. 3: indicate which GNSS station (e.g. VTE1? VTE2? etc.) is shown in the field photographs. I suggest to insert a new figure expanding the material of Fig.3 (lower right) showing better station baselines, faults, epicentres of October 26 and October 30, 2016 earthquakes and co-seismic ruptures. This will be a key figure to help readers to understand the relation between GNSS station location, earthquake effects etc.

Fig.4; there are five grey stars in this figure. Please leave only the 3 mainshocks and represent them with red colour. Clarify that InSAR scale bar is in cm.

Specific comments: line 3: replace discrete with campaign line 17: please indicate which agency provided moment magnitude estimation line 19: please indicate which agency provided moment magnitude estimation line 34: replace doy with DOY line 136: replace registrations with recordings

Reply: All the figures were reorganized according to your guidelines

Please also note the supplement to this comment:
http://www.nat-hazards-earth-syst-sci-discuss.net/nhess-2017-130/nhess-2017-130-AC2-supplement.pdf

---

## Author Response (AR1)

[revised manuscript text omitted]